# Discovering Native Ant Species with the Potential to Suppress Red Imported Fire Ants

**DOI:** 10.3390/insects15080582

**Published:** 2024-07-31

**Authors:** Meihong Ni, Xinyi Yang, Yiran Zheng, Yuan Wang, Mingxing Jiang

**Affiliations:** Ministry of Agriculture and Rural Affairs Key Laboratory of Agricultural Entomology, Key Laboratory of Biology of Crop Pathogens and Insects of Zhejiang Province, Institute of Insect Sciences, Zhejiang University, Hangzhou 310058, China; ni_meihong@zju.edu.cn (M.N.); xyyang1026@163.com (X.Y.); yiranzheng@zju.edu.cn (Y.Z.); 22016105@zju.edu.cn (Y.W.)

**Keywords:** interspecific interaction, aggression behavior, ant community, habitats, seasonal dynamics, spatial distribution

## Abstract

**Simple Summary:**

The red imported fire ant, *Solenopsis invicta* Buren, is a highly invasive species. Some native ants may show aggression towards *S. invicta* and thus suppress *S. invicta* invasion. However, only a few such native ants have been identified, and the available data are largely limited to their behaviors. Here, we not only observed the aggression levels between native ants and *S. invicta*, but also investigated their abundance in various habitats, the seasonal dynamics of their abundance, and their spatial distribution in habitats with *S. invicta*. Two ant species, *Monomorium chinense* and *Nylanderia bourbonica*, were demonstrated to show aggression towards *S. invicta* and could kill a large proportion of *S. invicta*. However, another ant, *Iridomyrmex anceps*, did not show aggression. *M. chinense* and *N. bourbonica* are more abundant in green belts (e.g., lawns) and grasslands relative to woodland and farmland. Their abundance is quite low in the early months of the year but increases in later parts of the season. *M. chinense* and *N. bourbonica* are restricted to marginal habitats, but after control of *S. invicta*, they can rebound rapidly within a few weeks. *M. chinense* and *N. bourbonica* have the potential to suppress *S. invicta* invasion in some habitats.

**Abstract:**

Native ants have long been considered for their potential to suppress the red imported fire ant, *Solenopsis invicta* Buren (Hymenoptera: Formicidae), a highly invasive and destructive species. However, the knowledge in this field is limited to behavioral observations of a few related native ants. In this study, by setting up a series of ant combinations of three native ants, i.e., *Monomorium chinense* Santschi, the robust crazy ant *Nylanderia bourbonica* Forel, and *Iridomyrmex anceps* Roger, with *S. invicta*, we observed the aggression levels and mortality rates. Using baited vials, we also investigated the abundance of native ants in four types of habitats in Eastern China that are preferred by *S. invicta* (woodland, green belts on roadsides, grassland, and farmland), as well as their seasonal abundance when co-existing with *S. invicta* and their spatial distribution before and after control of *S. invicta*. We found that *M. chinense* and *N. bourbonica* show a degree of aggression towards *S. invicta* and can kill substantial proportions of *S. invicta* under laboratory conditions, but *I. anceps* does not. Both *M. chinense* and *N. bourbonica* can occur in each type of habitat investigated and are more abundant in green belts (particularly lawns with turf) and grasslands relative to other habitats. In grasslands with *S. invicta*, *M. chinense* maintained a low density before early July; however, its abundance increased thereafter and reached a peak in September. *N. bourbonica* also had a low density early in the season and increased steadily from April. Its abundance began to decrease substantially from November. In grasslands invaded by *S. invicta*, both *M. chinense* and *N. bourbonica* were restricted to sites close to the margins before *S. invicta* was controlled; however, they spread to a larger range within a few weeks after control of *S. invicta*. In conclusion, *M. chinense* and *N. bourbonica* have the potential to suppress *S. invicta* invasion in habitats that are abundant with these two native ants.

## 1. Introduction

The red imported fire ant, *Solenopsis invicta* Buren (Hymenoptera: Formicidae), is one of the most widespread, abundant, and destructive invasive ants globally [1]. This ant has a broad range of negative, and often severe, impacts on agriculture, forestry, wildlife, infrastructure, and human health and lifestyle [2,3,4,5]. Native to South America [6], *S. invicta* was introduced to the United States between 1933 and 1945 and has since spread to the Caribbean, Oceania, Eastern Asia, and recently, to Europe [7,8]. In mainland China, since the first report of its presence in 2004 [9], *S. invicta* has spread to at least 620 counties or districts in 12 provinces [10] and has likely spread across a larger range due to human activities and climatic warming [11,12]. Various habitats, such as farmlands, forests, plant nurseries, grasslands, wetlands, and urban green spaces, can be infested by this ant [5,13], leading to serious economic and ecosystem service losses and biodiversity reduction in some regions [5]. Due to its multifaceted impacts, *S. invicta* is ranked as one of the most harmful invasive insect species in China, particularly in agro-ecosystems [14].

Native ants are generally vulnerable to *S. invicta* invasion and are often displaced if *S. invicta* thrives [2,5,15,16], primarily because of the high population density and the formidable competitive abilities of this ant [17]. However, some native ants can coexist with *S. invicta*, such as *Monomorium minimum* Buckley, *Paratrechina terricola* Buckley, and certain species of *Pheidole* and *Forelius* in the USA [18] and *Tapinoma melanocephalum* Fabricius, *Monomorium chinense* Santschi, and some species of *Pheidole* and *Paratrechina* in China [19,20,21]. Specifically, some (e.g., *M. minimum*) can invade *S. invicta* worker-defended colonies, prey upon broods, or prevent *S. invicta* workers from leaving the nest for foraging [22,23]. They may adopt certain protective tactics when encountering *S. invicta*, such as running away rapidly and releasing chemicals (venoms) that are deleterious to *S. invicta* [21]. For these reasons, a few native ants are believed to be capable of suppressing *S. invicta* (re)invasion to a certain degree, and it has been suggested that they be integrated into *S. invicta* control programs [22,23,24]. However, except for the few native ant species mentioned above, the potential that most native ants may hold for suppressing *S. invicta* remains unknown. As ants are one of the most diverse, abundant, and widespread animal groups in the world [25], some native ant species, undiscovered so far, are likely to be able to successfully persist with *S. invicta*.

With this in mind, it is crucial to learn where these ants reside and about their abundance in various habitats. Understanding their success in the habitats preferred by *S. invicta* would help us to evaluate their effectiveness within these habitats. However, such information is quite limited, even for the native ants that have been shown to be aggressive towards *S. invicta* (e.g., [23]).

In this study, we investigated the native ants that co-occur with *S. invicta* in Eastern China. This research was carried out in Eastern China because we had previously identified native ants in this region during our field investigations and during the implementation of *S. invicta* control programs. Moreover, Eastern China is one of the regions experiencing *S. invicta* expansion [10] and is being invaded by this ant on a larger scale [11,12]. Here, we first surveyed the aggressive interactions between three native ants and *S. invicta*. Then, we investigated their abundance in four types of habitats. Next, we performed a one-year investigation of the seasonal dynamics of native ants in a grassland, and their spatial distribution before and after the removal of *S. invicta* in the grassland. Our aims were to identify native ants that are aggressive towards *S. invicta* to determine their overall local distribution and population dynamics in the habitats that are preferred by *S. invicta* (i.e., grassland) and, based on these results, to evaluate their potential for suppressing *S. invicta* as a biotic factor to be included in *S. invicta* control programs.

## 2. Materials and Methods

### 2.1. Aggressive Interactions between Native Ants and S. invicta 

This experiment was performed by combining native ants with *S. invicta* in different ways. Levels of aggressive interactions were assayed, and the resulting levels of mortality were observed.

#### 2.1.1. Ant Collection

Based on our knowledge of the ant community in Eastern China, we selected three native ant species for aggression assays: *M. chinense*, the robust crazy ant *Nylanderia bourbonica* Forel, and *Iridomyrmex anceps* Roger. Their workers and *S. invicta* workers were collected using baited vials from one grassland and adjacent lawns located in Redwood Town in Longyou County, Zhejiang Province, China. The collected ants were kept in plastic containers (16 cm × 11 cm × 7 cm) coated with talcum powder and maintained in a climate-controlled chamber (25 °C ± 2 °C and 75% ± 10% RH with a photoperiod of 14:10 L:D). The ants were fed ham sausage (Wangzhongwang, Shuanghui Industry Group Co., Ltd., Luohe, Henan Province, China; the same below). They were used for assays within 24 h after collection. The assays were performed under laboratory conditions at temperatures of 25–27 °C.

#### 2.1.2. Dyad Interactions

As differently sized ants may show different interaction outcomes [26], we used large-, medium-, and small-sized *S. invicta* workers (head width > 1.35 mm, 1.00–1.35 mm, and <1.00 mm, respectively) separately for the assays, with each being paired with workers of each of the three native ants mentioned above. Within each native ant species, all of the workers had a similar body size: *M. chinense*, 1.3–1.7 mm; *N. bourbonica*, 2.3–3.3 mm; and *I. anceps*, 3.0–4.0 mm. The assay methods were similar to those described by Buczkowski and Bennett [27]. Briefly, we placed one native ant and one *S. invicta* in a 1.5 mL Axygen centrifuge tube, immediately settled the tube horizontally, and started to observe their behaviors. The observation lasted 5 min for each ant pair, during which we recorded the timepoint of the ants initiating physical contact, the duration of the contact, and the aggression score (levels) of the contact. The aggression scores were determined by referring to Grangier et al. [28] with some modifications, including 1. ants make short (≤1 s) antennal contact; 2. ants make long (>1 s) antennal contact; 3. one or both ants in the pair exhibit threatening postures by opening the mandibles or curling the gaster, or bite quickly; and 4. the ants fight by prolonged biting and eventually stinging. The score was recorded as “0” if either ant alone or both avoided contacting the other throughout the trial. As one trial ended, the maximum score (i.e., a number from 0 to 4) attained in the trial was determined. Thirty trials (replicates) were performed for each type of native ant and *S. invicta* combination. The effects of *S. invicta* body size on the aggression scores were analyzed; the means were compared between the combinations with different *S. invicta* sizes and between the combinations with different native ants.

#### 2.1.3. Symmetrical Group-on-Group Interactions

For each native ant species, 1, 3, 5, and 10 workers were released separately into talcum-powder-coated plastic containers (4.2 cm in diameter, 6.8 cm high). After 5 min, the same number of medium-sized *S. invicta* workers were introduced to the containers, forming four types of ant combinations: 1 vs. 1, 3 vs. 3, 5 vs. 5, and 10 vs. 10. Here we elected to use medium-sized *S. invicta* workers because they are the primary force for defense, carrying larger venom arsenals, longer stings than small-sized workers [29], and greater in number than large-sized workers. At 12, 24, and 36 h, the numbers of dead ants (without visible movement of the legs) or dying ants (unable to stand) were recorded for each partnership. We performed 30 trials for each combination, taking every 10 trials as one replicate (i.e., there were three replicates in total). Within each replicate, the number of dead/dying ants were pooled for each partnership, and their percentages (mortality rates) in the pooled initial number of introduced ants (i.e., 10, 30, 50, and 100 ants) were calculated. The mortality rates of the native ants and *S. invicta* at 36 h after treatment initiation were used to analyze the relations with the ant combination (factor 1) and species of native ant (factor 2).

#### 2.1.4. Asymmetrical Group-on-One Interactions

All the methods of this assay were similar to those described above for the symmetrical group-on-group interactions, except that only one medium-sized *S. invicta* worker was used in each combination. Additionally, the combination with 10 native ants was set up for *M. chinense* due to its smaller size to balance the biomass disparity with the *S. invicta* worker by providing a greater number of *M. chinense*.

### 2.2. Abundance of Native Ants in Different Types of Habitats

#### 2.2.1. Location and Habitats

The location of this investigation was in Redwood Town and an adjacent 1.3 km × 1.1 km region (29.072–29.084° N; 119.229–119.254° E) where *S. invicta* was introduced unintentionally before 2016 and was chemically controlled by the local government from late August 2019 to March 2020. The control consisted of two steps. Applying Kai Rui^®^ (baits, 0.73% hydramethylnon, BASF, Ludwigshafen, Germany) first to the margin of *S. invicta* mounds at a dosage of 20 g per mound, after 3–4 days a mixture of 5% imidacloprid and 40% chlorpyrifos was poured into nests. This method can reduce the contact of native ants to chemicals and, thus, be perceived to be safe to them. Our investigations were performed in four types of habitats, namely woodland, green belt on the roadside, grassland, and farmland, after the removal of *S. invicta* in the corresponding habitats. To reduce unexpected impacts from *S. invicta*, for each type of habitat, only the sites with a previous infestation of 225–300 *S. invicta* nests per ha were selected. No active *S. invicta* mounds were found in the selected sites.

The selected woodland was located on a hill approximately 1 km in perimeter and approximately 50 m in altitude. The trees in this woodland were mostly <4 m in height, spaced 3–6 m apart, and dominated by *Cinnamomum camphora* (L.) Presl. and *Ilex chinensis* Sims. Abundant natural shrubs and perennial grasses were present as ground-cover vegetation. The green belt on the roadside neighboring the hill was 3–8 m in width and approximately 300 m in length. The vegetation in the belt was dominated by *Cynodon dactylon* (L.) Pars. The belt was decorated with rows of shrubs (e.g., *Buxus megistophylla* H. Lév.). For the grassland habitat, which was scattered within the study location, three sites were selected for investigation, with one (approximately 28 ha) at the southeastern margin of the town, one (10 m × 800 m) adjacent to local farmland, and one (9.24 ha) surrounded by roads and factory buildings, with each dominated by *Imperata cylindrica* (L.) Beauv. as ground vegetation. The farmland (50 m × 600 m) was approximately 130 m to the north of the hill described above and was previously used to grow rice. It had been abandoned for at least two years prior to our study. 

#### 2.2.2. Ant Sampling

For each type of habitat, sampling began approximately seven months after the control of *S. invicta*, i.e., from June to October 2020, except for one grassland site, where sampling began on 16 September 2019, nearly three weeks after *S. invicta* control had been carried out. By July 2022, four to six samplings had been obtained for each type of habitat, occurring from June to October. Vials (25 mL) baited with a piece of ham sausage (approximately 1 g) were used for the sampling, and 20–185 such vials were used at each site, depending on the size of the sampled sites. The baited vials were placed on the ground at a space of 10 m and were collected after 30 min by quickly capping them to trap any ants inside. To ensure baiting efficiency, the samplings were performed between 1600 and 1800 h, when the ants were speculated to be actively foraging.

#### 2.2.3. Ant Identification and Sorting

The collected ants were identified by referring to the ants’ morphological features, as described by Wu and Wang [30], Zhou [31], and Zhou and Chen [32]. Some ants were identified as belonging to genera only due to a lack of available references. The identification of species/genera was confirmed by cloning a 710 bp fragment of ant mitochondrial *COI* [33], followed by sequencing (Tsingke Biotech, Hangzhou, China) and blasting the sequence in the National Center for Biotechnology Information (NCBI) database. The sequences have been submitted to GenBank, as shown in Appendix A. The number of vials used for trapping the ants (with at least one ant) and the number of ants in each vial were recorded for each batch (one batch was defined as the sampling performed in one type of habitat on one sampling date), and their corresponding percentages in the batch were calculated. Using these percentages, the abundance of native ant species was compared between habitat types and between sampling dates to identify the differential distribution of the major native ants in the local ecosystem, particularly those aggressive towards *S. invicta*.

### 2.3. Seasonal Dynamics of Native Ants Coexisting with S. invicta

After finding that grassland is one of the major habitats of *M. chinense* and *N. bourbonica*, the two native ants that are aggressive towards *S. invicta*, we further investigated their seasonal dynamics in one *S. invicta*-invaded grassland plot (ca. 10 m wide, 460 m long) located in the middle of Zhejiang Province, adjacent to the east of Jinhua New Energy Vehicle Town. The plot had a north–south orientation, was neighbored in the west by a ditch (ca. 2 m width), and was neighbored in the east by a few ponds, cultivated farmlands, and abandoned farmlands. There were 40–70 *S. invicta* mounds on the plot (87–152 mounds per ha.), which varied with the season. The vegetation was at a medium density, dominated by the Poaceae and Asteraceae families, and at heights of mostly <80 cm. The investigation began in mid-February 2023 and ended in mid-February 2024. Ant sampling was performed every 7–10 d from mid-February to mid-April, monthly from May to October 2023, twice in November 2023, and once in mid-February 2024. No sampling was performed from December 2023 to January 2024, which was the winter season. The investigation was not replicated in this plot in 2024, as *S. invicta* had to be controlled in early March 2024 in an eradication program of this pest by the local government.

We obtained a total of 18 ant samplings in the plot, adopting the sampling method described above for the investigations in different habitats. For each sampling, 46 vials baited with ham sausage were used, placed every 10 m, and retrieved after 30 min. *S. invicta*, *M. chinense*, *N. bourbonica*, and other ant species in the vials were counted, and the percentages of vials with trapped ants were calculated with respect to each native ant species.

The percentage of baits that trapped *N. bourbonica* increased steadily from early April to early June 2023, and the number of this ant reached >50 in a few vials (see Results). We considered it likely that this trend might also occur in other grasslands, representing a typical population feature of *N. bourbonica* under local conditions. To clarify this, we selected another grassland plot (ca. 20 m wide, 160 m long; *S. invicta* density ca. 88–128 mounds per ha.) on the western side of the first grassland plot and performed the same investigation from 10 June 2023 to 30 December 2023. The soil type of this plot was mineral, and the vegetation was also dominated by the Poaceae and Asteraceae families, with a much higher plant density. Eight ant samplings were obtained, and for each sampling, 30 baited vials were used, which were arranged in two rows with 15 vials in each row.

In addition, we also investigated the native ants in nine locations in Zhejiang and Jiangxi provinces, where measures had not been taken to control *S. invicta* or control measures had been taken but this ant was not absolutely removed. We selected these locations randomly, with the aim of collecting supporting information showing that *M. chinense* and *N. bourbonica* can coexist with *S. invicta* in different regions and still increase their population later in the season, e.g., from August to October, as shown in previous investigations.

### 2.4. Spatial Distribution of Native Ants Coexisting with S. invicta

Alongside the abundance (in different habitats) and seasonal dynamics (in major habitats), the spatial distribution of native ants within *S. invicta*-invaded habitats can also be quite informative regarding their ecological responses toward *S. invicta*. In 2021, we investigated this on the banks of the Dongyangjiang River (N 119°74′, E 29°09′) in mid-Zhejiang. The banks are trapezoid shaped in cross-section, 600 m long in the east–west orientation, with one road of ca. 4 m width at its top. On the northern and southern sides of the road, there is one slope each, which are ca. 18 m and 8 m in width, respectively. The dominant vegetation is turf and grasses of various families. Samplings of ants were performed on three dates, namely 3 September, 18 September, and 9 October, i.e., 1 d before, 2 weeks after, and 5 weeks after control of *S. invicta*, respectively. The control of *S. invicta* was carried out on 4 September using the same chemical method stated in Section 2.2.1. On each sampling date, 360 baited vials were used for trapping ants, with 180 vials for each slope, which were arranged in three rows (60 vials per row): one row on the upper side of the slope, one row on the lower side of the slope, and one in the middle. Within each row, the vials were placed on the ground every ca. 7.5 m. The other methods were the same as described above.

### 2.5. Data Analysis

SPSS 25 [34] was used for the data analysis. The Kolmogorov–Smirnov test and Levene’s test were used to test the normality and the variance homogeneity of the aggression score data, respectively. For data with a normal distribution and equal variance, a one-way analysis of variance (ANOVA) was performed to analyze the effect of *S. invicta* body size and native ant species on the aggression level, and the means were compared using Tukey’s multiple comparison test. For aggression score data with a non-normal distribution or without equal variance, the Kruskal–Wallis *H* test was performed, and means were compared using the Mann–Whitney *U* test. The same methods were used to analyze data on ant mortality from group-on-group and group-on-one interactions to discover relations of mortality with the number of ants (ant combination) and the species of native ants. The significance level was 0.05.

For the proportion data of baits trapping ants 1 d before, 2 weeks after, and 5 weeks after *S. invicta* control, chi-square tests were used to analyze the differences among these dates. For the number data of trapped ants, an ANOVA was used to test the associations with sampling date, and Tukey’s test was used to compare means. The number data were square-root transformed prior to analysis. The significance level was 0.05.

## 3. Results

### 3.1. Aggression Interactions between Native Ants and S. invicta 

#### 3.1.1. Dyad Interactions

*S. invicta* body size had a significant effect on the scores of the aggression interactions between *S. invicta* and *N. bourbonica* (*H* = 8.536, df = 2, *p* = 0.014; Kruskal–Wallis *H* test) and between *S. invicta* and *M. chinense* (*H* = 6.327, df = 2, *p* = 0.042) but did not for interactions between *S. invicta* and *I. anceps* (*H* = 3.661, df = 2, *p* = 0.160). In pairs with *N. bourbonica* or *I. anceps*, the highest score was observed for those interacting with large-sized *S. invicta*, reaching 3.45 and 3.40, respectively, and the score declined with the size of *S. invicta*. However, in pairs with *M. chinense*, the highest score (2.70) was observed in those interacting with medium-sized *S. invicta* (Table 1). Overall, pairs with *N. bourbonica* and those with *I. anceps* had similar scores, irrespective of *S. invicta* size, and they had higher scores than those with *M. chinense*, which statistically reached a significant level when interacting with large or small *S. invicta* (*p* < 0.05, Mann–Whitney *U* test; Table 1).

When meeting *S. invicta*, *N. bourbonica* typically run rapidly, exhibiting few avoidance or defensive behaviors. They showed on average only 1–2 aggressive acts of level 3 (threatening or biting quickly) and level 4 (fighting) (Figure 1A) evoked by *S. invicta*. Fights between *N. bourbonica* and small-sized *S. invicta* lasted for 16 s on average (Figure 1B). However, with medium-sized *S. invicta*, the fight lengths shortened to only approximately 2 s, and the threatening (of *S. invicta*) extended to 13.5 s (Figure 1B). When interacting with large-sized *S. invicta*, both fights and threatening were shortened to less than 5 s. Aggressive acts (levels 1–4) were mostly initiated within 100 s after initiation of treatment, irrespective of the body size of *S. invicta* (Figure 2).

Notably, most *S. invicta* avoided *M. chinense*, although the latter is much smaller in body size and is evidently slower than the former. We observed very few interactive acts of levels 2–4 (i.e., from making >1 s antennal contacts to fighting) in pairs with large- or small-sized *S. invicta*; the number of acts increased a little with medium-sized *S. invicta* (Figure 1A). Most of the observed acts lasted <5 s, and a few acts extended to nearly 8 s with large-sized *S. invicta* (Figure 1B). The time for initiation of acts was variable and often late (Figure 2).

*I. anceps* generally moved around rapidly and avoided contact with *S. invicta*, particularly when paired with large- or medium-sized *S. invicta*. In contrast, *S. invicta* typically exhibited a threatening posture (level 3) toward *I. anceps* 2.5–3.7 times on average in 5 min (Figure 1A), and the threatening behavior lasted much longer (>30 s) when the *S. invicta* was medium or small in size (Figure 1B). In such pairs, fights (level 4) could also take place, which lasted for 7–11s on average. Most acts were initiated within 100 s, irrespective of the body size of *S. invicta* (Figure 2).

#### 3.1.2. Symmetrical Group-on-Group Interactions

In most cases, the mortality rates of the native ants and *S. invicta* were related significantly or nearly significantly to the number of ants and the species of the native ant under observation (Table 2). With *M. chinense*, *S. invicta* suffered higher mortality than *M. chinense* (Figure 3A), although *S. invicta* initiated nearly all of the aggressive acts. Strikingly, in the 5 vs. 5 and 10 vs. 10 combinations with *M. chinense*, approximately 65% and 80% of *S. invicta* died (including dying individuals, the same below) within 36 h, which was a significantly rate higher than *M. chinense* (5 vs. 5: *Z* = −1.993, *p* = 0.046; 10 vs. 10: *Z* = −1.993, *p* = 0.046). In combinations of *S. invicta* and *N. bourbonica*, over half of both died in some trials, and their mortality rates did not differ significantly (*p* > 0.05), except in the 1 vs. 1 combination at 36 h and the 10 vs. 10 combination at 12 h, where significantly more *N. bourbonica* died (*Z* = −2.023, *p* = 0.043; and *Z* = −1.964, *p* = 0.050, respectively; Figure 3B). *I. anceps* died at higher proportions than *S. invicta,* as their number increased to 5–10, particularly in 10 vs. 10, where nearly 75% of *I. anceps* but only 15% of *S. invicta* had died by 36 h (Figure 3C).

#### 3.1.3. Asymmetrical Group-on-One Interactions

The cumulative mortality of each native ant species was mostly below 30% at 36 h after treatment (Figure 4A). The mortality of *M. chinense* related significantly with the number of ants used (*p* < 0.05; Table 3); nearly one-third of this type of ant died in the 1 vs. 1 and 5 vs. 1 trials, which was significantly higher than the mortality in the 3 vs. 1 and 10 vs. 1 trials (13–15%; *p* < 0.05). This relation was nearly significant for *N. bourbonica* and *I. anceps* (*p* = 0.059, and *p* = 0.057, respectively; Table 3). Nearly 70% of *N. bourbonica* were killed in the 1 vs. 1 combination (Figure 4A).

The mortality rate of *S. invicta* was significantly related to the number of *N. bourbonica*, but not to the number of *M. chinense* and *I. anceps* (Table 3). In trials with five *N. bourbonica* or three *M. chinense*, nearly 75% and 100% of *S. invicta* were killed within 36 h, respectively, which was significantly higher than the corresponding native ant (24% and 13%, respectively; *p* < 0.05). Considerable *S. invicta* mortality was also observed (27–60% within 36 h) in all of the other trials using *N. bourbonica* and *M. chinense*. In contrast, in all of the combinations with *I. anceps*, <10% of *S. invicta* died (Figure 4B).

Both at 1 vs. 1 and 3 vs. 1, the mortality of the native ants depended significantly on their species (*p* < 0.05). More *N. bourbonica* were killed than the other two native ants (Figure 4A). The relation with native ant species was not significant for 5 vs. 1 (*p* = 0.223; Table 3). For each combination, the mortality of *S. invicta* was significantly related to the species of native ant (*p* < 0.05; Table 3). More *S. invicta* were killed, as mentioned above, by *N. bourbonica* at 5 vs. 1 and by *M. chinense* at 3 vs. 1, while fewer were killed by *I. anceps* in all combinations.

### 3.2. Abundance of Native Ants in Various Habitats

Using baited vials, we collected a total of 17 batches with 121,139 native ants from the woodlands, road green belts on roadsides, grasslands, and farmland. They belonged to 10 species and three subfamilies, Myrmicinae, Dolichoderinae, and Formicinae (Table 4). *M. chinense*, *N. bourbonica*, and *I. anceps* were captured in 27.9%, 24.8%, and 19.5% of the 1590 baits that trapped ants, accounting for 41.1%, 10.9%, and 16.6% of the total ants trapped, respectively. We also found many other native ants, including *Tetramorium caespitum* L., *Ochetellus glaber* Mayr, and one *Pheidole* species, which were captured by 12.3%, 9.4%, and 5.0% of the baits, respectively, and *Nylanderia flavipes* Smith, *Pristomyrmex punctatus* Smith, another *Pheidole* species, and one *Crematogaster* species, each captured by <1% of the baits (Table 4).

Within batches, *M. chinense*, *T. caespitum*, *N. bourbonica*, and *I. anceps* were more abundant than the other ants, which accounted for 30.7%, 21.0%, 18.0%, and 15.7%, on average, of the total number of ants trapped, and 23.0%, 13.3%, 25.6%, and 11.6% of the total number of baits that trapped ants, respectively. More *M. chinense* were found in the green belt on roadsides, reaching >40 individuals per bait in one-half (3/6) of the batches (Figure 5A). In contrast, both *N. bourbonica* and *I. anceps* tended to be more abundant in woodlands and grasslands (and *I. anceps* was rarely found in the road green belt), and *T. caespitum* occurred dominantly in farmland (Figure 5A,B). When comparing between batches, *M. chinense* and *N. bourbonica* were more abundant in the later batches relative to the earlier batches (Figure 5A).

### 3.3. Seasonal Dynamics of Native Ants Coexisting with S. invicta

From 18 February 2023 to 18 February 2024, we trapped a large number of *S. invicta* in the investigated grassland plot (#1) in at least 40% of the baited traps (Figure 6A,B; no ants were trapped on 11 February 2023 due to low winter temperatures). Both *M. chinense* and *N. bourbonica* were trapped on at least two-thirds of the 18 sampling dates. *M. chinense* was not abundant before early July, and its peak occurred from mid-August to early October, when it was trapped by 30% of the baits. *N. bourbonica* was not abundant before mid-April (trapped by <10% baits), while it increased steadily after this, becoming trapped in ca. 20% of baits from July to October. Very low *N. bourbonica* trapping rates occurred on 11 September for unknown reasons. Both *M. chinense* and *N. bourbonica* decreased from early November (Figure 6A,B).

In the other grassland plot (#2), *S. invicta* was trapped by 33–63% of the baits from early June to late November. Few *M. chinense* were trapped. *N. bourbonica* were trapped at much higher rates, occurring in nearly 20% of the baits during October and early November and decreasing from late November (Figure 6C). Combined with the results from plot #1, we found that both *M. chinense* and *N. bourbonica* may vary greatly between different grasslands in their abundance.

Both *M. chinense* and *N. bourbonica* were found in other locations, again with great variance in their abundance, and each could coexist with *S. invicta* (Table 5). Even at locations heavily invaded by *S. invicta* (trapped by 100% of baits), *M. chinense* and *N. bourbonica* could be present and could occur simultaneously with *S. invicta* in the same baits. These two native ants could also occur simultaneously in the same baits.

### 3.4. Spatial Distribution of Native Ants Coexisting with S. invicta

We found six native ant species in the grassland plot present with *S. Invicta*, namely *M. chinense*, *N. bourbonica*, *O. glaber*, *T. caespitum*, *Pheidole* sp., and *Solenopsis* sp., with the former two species being dominant. Before the control of *S. invicta*, most native ants were restricted to sites close to the margins of the investigated plots, and a majority of *M. chinense* and *N. bourbonica* were distributed in only two patches (indicated by the dashed rectangle in Figure 7A). However, 2 weeks and 5 weeks after *S. invicta* control, both *M. chinense* and *N. bourbonica* spread to a larger range (Figure 7B,C). Particularly, *M. chinense* was trapped by 13.6% of the baits 5 weeks after control, which was significantly higher (5.8%) than before control (χ^2^ = 12.406, df = 1, *p* < 0.001, chi-square test) and the percentage (8.9%) at 2 weeks after *S. invicta* removal (χ^2^ = 4.020, df = 1, *p* = 0.029; Figure 8A). Moreover, the number of *M. chinense* trapped per bait did not decline significantly despite the spatial dilution of the population (Figure 8B).

## 4. Discussion

Native ants have long attracted attention for their potential role in *S. invicta* control, but available data are largely limited to the behavioral performances of related ants, with little regarding their distribution and seasonal dynamics. Here, we not only identified the ants that show aggression towards *S. invicta*, but also determined the type of habitats abundant with them, their seasonal abundance in the preferred habitat, and their spatial distribution before and after *S. invicta* removal. This, to our knowledge, is one of the few comprehensive studies in this field, and the first report in Eastern China, a region heavily invaded by *S. invicta* in some habitats.

We showed that, as *M. chinense*, *N. bourbonica,* and *I. anceps* encounter *S. invicta* in a limited space, threatening and fights may take place (Table 1; Figure 1). As a result, a substantial proportion of *S. invicta* can be killed by *M. chinense* and *N. bourbonica* (Figure 3A,B and Figure 4B). Moreover, *M. chinense* and *N. bourbonica* are abundant in some habitats (Table 4; Figure 5), where they are able to maintain populations instead of being displaced by *S. invicta* (Figure 6 and Figure 7). This suggests that, if *S. invicta* arrives at sites it prefers (e.g., grassland and lawn) that are abundant with *M. chinense* and *N. bourbonica*, its establishment and subsequent local expansion might be delayed or reduced.

*M. chinense* showed impressive aggression during this study, as irrespective of its much smaller body size and lower movement speed relative to *S. invicta*, it often fought with large *S. invicta* (lasting for 8 s, as observed in dyad interactions, Figure 1B) and killed more *S. invicta* than itself was killed by *S. invicta* (Figure 3A). *M. chinense* needed only three individuals to kill one *S. invicta* (Figure 4B). This result was in accordance with Chen et al. [23], who reported that *M. chinense* can kill *S. invicta* through chemical defense (venom). Our experimental design differed from Chen et al. [23]. We used combinations of different numbers of ants, allowing us to observe more aggression-related detail in the behaviors, whereas Chen et al. [21] used an equal biomass of the two ants. Combining the results from other reports (e.g., [21]), it appears that *M. chinense* shows aggression or defensiveness as a crucial tactic to cope with *S. invicta*. Interestingly, similar findings have also been reported for *M. chinense* when encountering the Argentine ant, *Linepithema humile* Mayr, another invasive ant species [35]. Thus, the success of *M. chinense* in habitats invaded by alien ants might be a common, if not ubiquitous, phenomenon.

The behavior of *M. chinense* described above might be enhanced in groups (this is often observed in this ant, possibly due to intraspecific cooperation). As indicated in the trials of the 10 vs. 10 combination, where *M. chinense* often stayed in groups with little movement, *S. invicta* suffered a higher mortality relative to the combinations with fewer ants, i.e., 5 vs. 5 (Figure 3A). On the other hand, the staying-together behavior of *M. chinense* might induce certain responses in *S. invicta*. For instance, in the survey with three (fewer) *M. chinense* and one *S. invicta*, the *S. invicta* ant may have chosen to attack more often, thereby suffering a higher mortality rate. In contrast, in the surveys with five or ten (more) *M. chinense* and one *S. invicta*, the *S. invicta* possibly reduced its attacks and thus suffered a lower mortality rate (Figure 4B). In other words, the outcomes of *M. chinense* and *S. invicta* interactions were probably highly density dependent.

*N. bourbonica* occurs in various habitats, particularly woodlands and grasslands (Figure 5), and is often found in the vicinity and even on the surface of *S. invicta* mounds (based on observation). This is in line with previous reports that *N. bourbonica* may be a dominant species in local ant communities invaded by *S. invicta* [36,37]. In fact, *N. bourbonica*, native to Southeastern Asia and introduced to other regions (such as parts of North America and the West Indies [36]) is an invasive species itself and is considered a widespread household and agricultural pest in some regions [38,39,40]. There are several factors related to its success, for example, the ability to adapt to frequent disturbance and marginal habitats [41] and the use of chemicals when fighting with other ants (based on observation). However, *N. bourbonica* has also been reported to be a weak competitor to *S. invicta* in some regions [37]. Thus, the performances of *N. bourbonica*, and its interactions with *S. invicta*, may differ in different habitats or regions. Other species of *Nylanderia*, some of which have been reported to be able to take advantage of spaces not yet occupied by *S. invicta* [42], were not found in our study.

*I. anceps* showed no aggression and was sensitive to *S. invicta* attack (Figure 3C). This was consistent with our observations that *I. anceps* rarely occurs in the vicinity of *S. invicta* mounds (based on observation), and with another report that *Iridomyrmex* spp. generally occurs in lower numbers in the areas invaded by invasive ants, e.g., *Anoplolepis gracilipes* Smith, *L. humile*, and *Pheidole megacephala* Fabricius [15]. However, some *Iridomyrmex* species have been documented as having a strong competitive ability, and regions rich with them may be resistant to ant invasions ([1], and the references therein). In Australia, for example, *Iridomyrmex* spp. are thought to have limited the spread of invasive ants [43,44,45] due to their widespread distribution, high abundance, activity levels, and behavioral aggression [46]. Some *Iridomyrmex* ants can recover rapidly after the removal of *S. invicta* [47]. Thus, considering that *Iridomyrmex* often dominate warm, open habitats in terms of both number and function ([43] and this study), some of its species may have the potential to suppress *S. invicta* invasion, although this has not been found yet in China.

All of the other native ants that we found, including *T. caespitum*, *O. glaber*, and the two *Pheidole* species, did not show evident aggression (unpublished data). Despite this, attention needs to be given to *Pheidole* ants, as some have been reported to be very aggressive, and even to kill or prey upon invasive ants [21,48,49], or show superiority in exploitative competition over invasive ants [50]. In addition, some *Pheidole* species are ubiquitous in the regions invaded by *S. invicta* or other ants [51,52,53,54]. Thus, *Pheidole* ants of interest could exist elsewhere.

Native ants were present near the margins of plots before *S. invicta* control (Figure 7), and some (e.g., *M. chinense*) rebounded rapidly after control (Figure 8A), implying that marginal areas may serve as a refuge for native ants. We suggest giving more research focus to habitat margins for their significance to native ants. Moreover, the heterogeneity within habitats should also be analyzed because some of the variables could have important effects on native ants’ abundance, including disturbance, micro-climate, soil type, vegetation cover, and the partition of food resources [55,56,57,58,59].

It is well known that both *M. chinense* and *N. bourbonica* have a broad distribution range in China. They exist in (at least) Eastern, Middle, and Southern China [30,59,60], and specifically, *M. chinense* has been reported to coexist with *S. invicta* in different regions [20,23,61,62] and habitats ([63] and this study). Thus, this could be an example of the coexistence of native ants with invasive ants, which has been reported previously at the regional scale [64].

In conclusion, we identified two native ants, *M. chinense* and *N. bourbonica*, that show aggression towards *S. invicta* and possess the potential to suppress *S. invicta* invasion. They could be integrated with supplemental control methods that are under development, such as botanical insecticides and pathogens [65,66,67]. Our study is one of the few involving the interspecific interactions of native ants with *S. invicta*, and we hope our results will enrich the knowledge in this field. Further research is needed to understand the behavior of *M. chinense* and *N. bourbonica* at the colony level, their dynamics during different stages of *S. invicta* invasion, and as mentioned above, the effects of within-habitat factors. Future research may focus on grasslands and lawns, which are favored by *S. invicta* and which probably contain an abundance of *M. chinense* and/or *N. bourbonica*.

## Figures and Tables

**Figure 1 insects-15-00582-f001:**
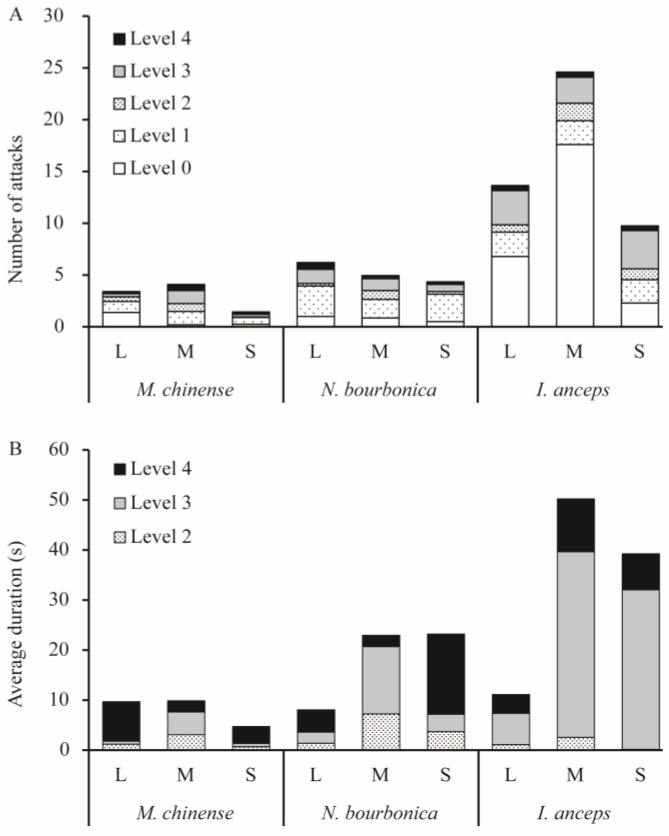
Number of each level of aggression act (**A**) and the average duration of levels 2–4 (**B**) observed in dyad interactions between *Solenopsis invicta* and native ants. Level 0—avoidance, level 1—ants make short (≤1 s) antennal contact, level 2—ants make long (>1 s) antennal contact, level 3—ants exhibit threatening postures or bite quickly, and level 4—fighting.

**Figure 2 insects-15-00582-f002:**
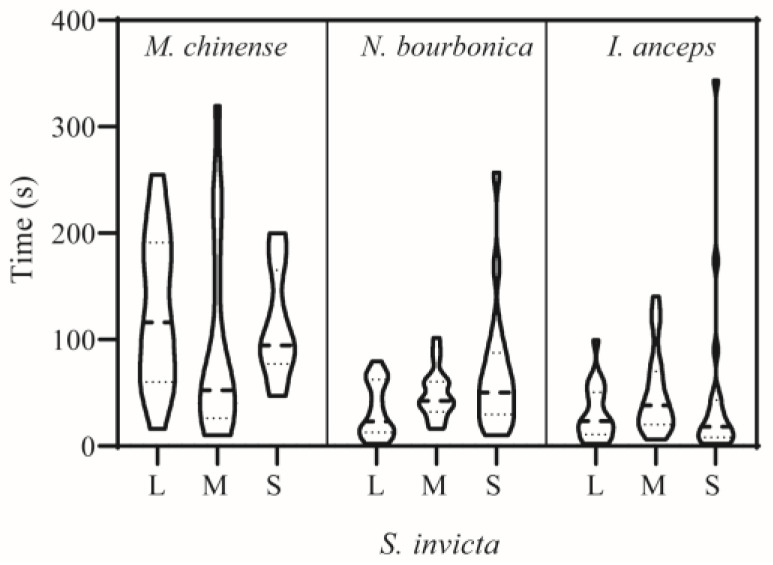
Time of the first attack in dyad interactions between *Solenopsis invicta* and native ants. L, M, and S stand for large-, medium-, and small-sized *S. invicta* used in the trials, respectively. Refer to the note in Figure 1 for information on aggression levels. All repetitions are represented for all ant pairs displaying aggressive acts from levels 1–4.

**Figure 3 insects-15-00582-f003:**
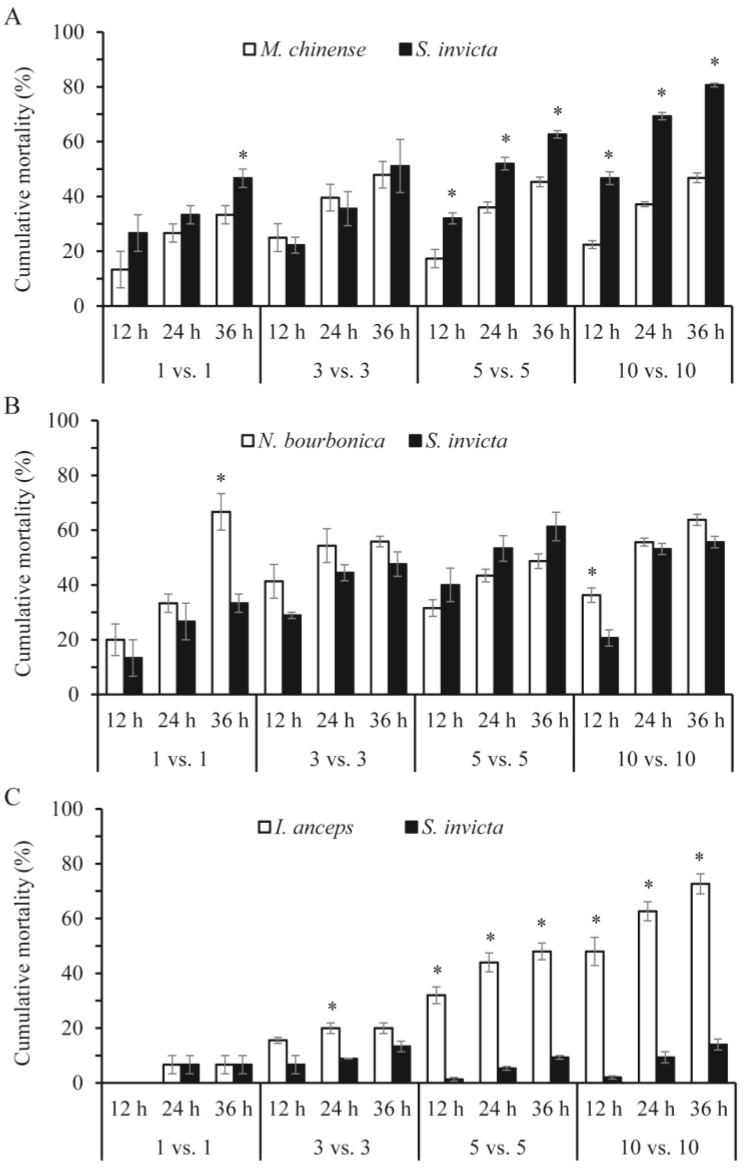
Cumulative mortality of native ants and *Solenopsis invicta* at 12, 14, and 36 h in the group-on-group interaction tests, where groups of 1, 3, 5, and 10 native ants were placed with the same number of *S. invicta*. (**A**), *Monomorium chinense* + *S. invicta*; (**B**), *Nylanderia bourbonica* + *S. invicta*; and (**C**), *Iridomyrmex anceps* + *S. invicta*. The asterisk indicates a significant difference in mortality between the native ants and *S. invicta* (*p* < 0.05, Mann–Whitney *U* test).

**Figure 4 insects-15-00582-f004:**
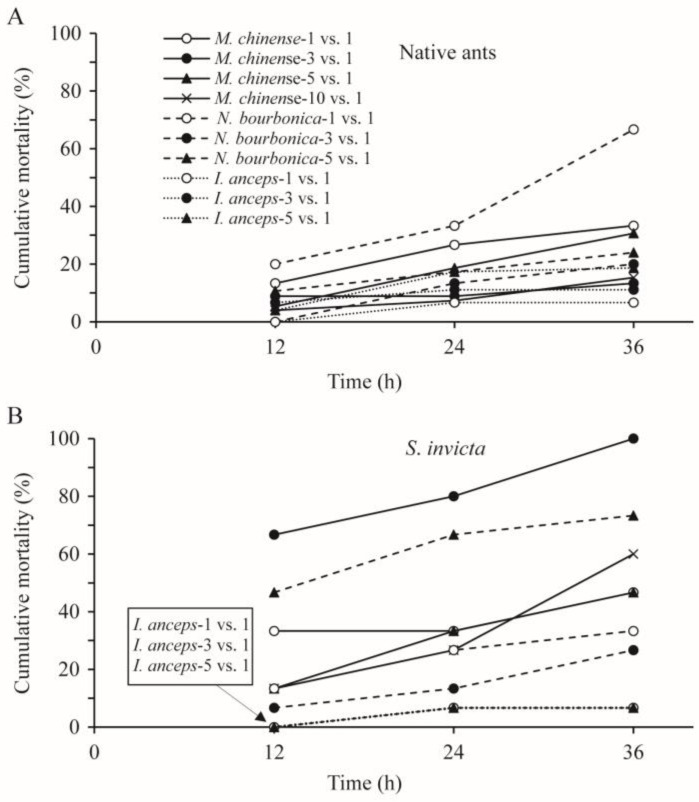
Cumulative mortality of native ants (**A**) and *Solenopsis invicta* (**B**) at 12, 14, and 36 h in the group-on-one interaction tests, where groups of 1, 3, 5, and 10 native ants were placed independently with an individual *S. invicta*.

**Figure 5 insects-15-00582-f005:**
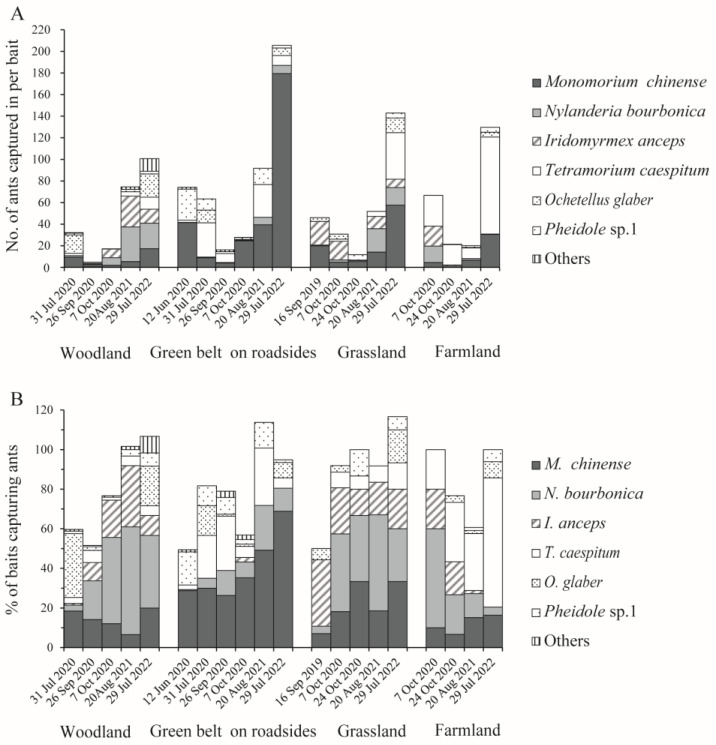
The average number of native ants trapped per bait (**A**) and the percentage of baits that trapped them (**B**) in the four types of habitats under investigation.

**Figure 6 insects-15-00582-f006:**
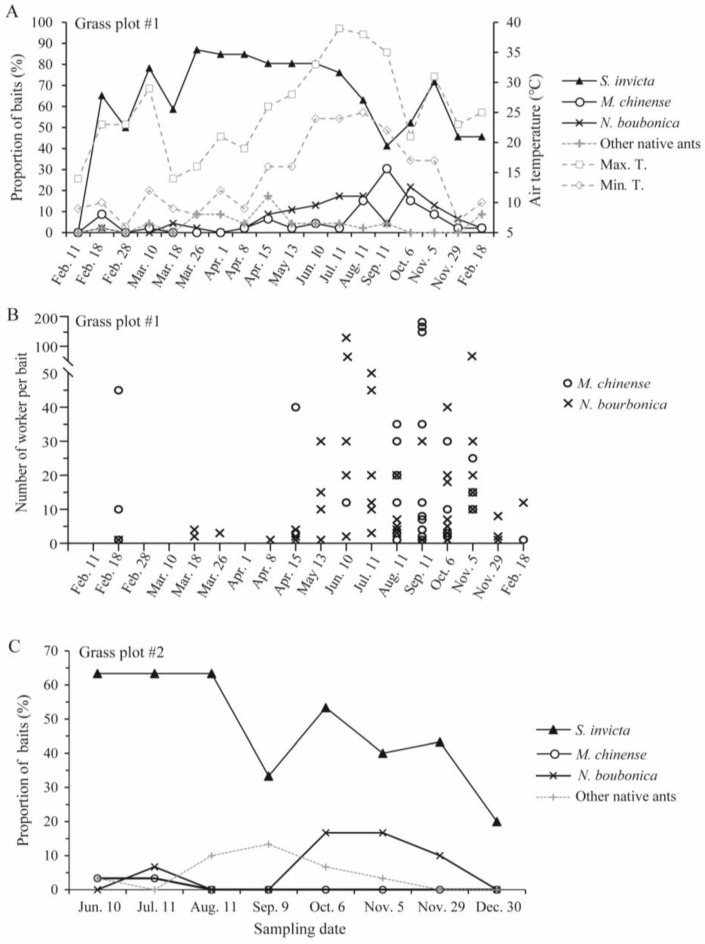
Seasonal dynamics of native ants coexisting with *S. invicta* in two grass plots near Jinhua New Energy Vehicle Town, as revealed by the proportions of baits trapping different ants (**A**,**C**) and the number of the two native ants (*M. chinense* and *N. bourbonica*) in the vials in one grass plot (**B**).

**Figure 7 insects-15-00582-f007:**
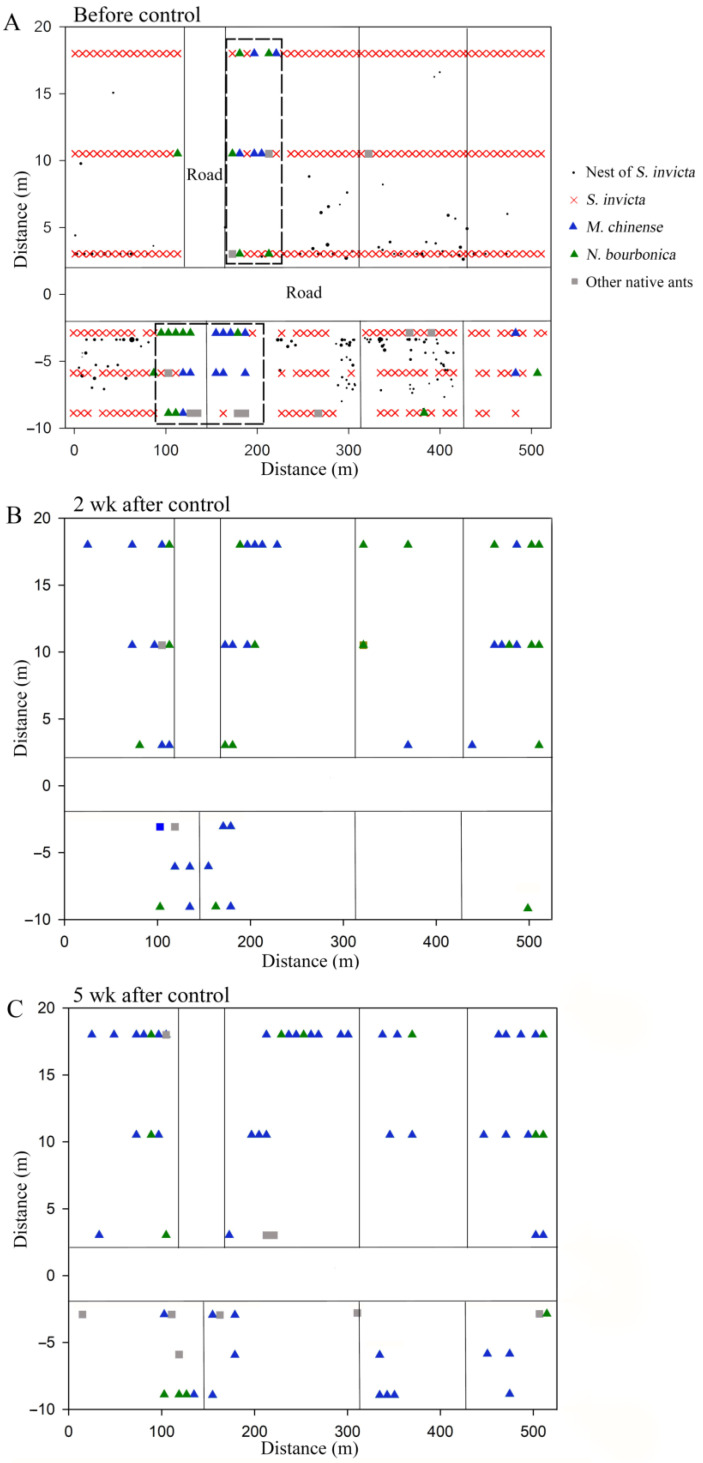
Spatial distribution of native ant species before control of *S. invicta* (**A**) and 2 weeks (**B**) and 5 weeks (**C**) after control of *S. invicta* in the grasslands on the banks of the Dongyangjiang River in Jinghua, Zhejiang province. The dashed rectangle in (**A**) indicates the major patches with *M. chinense* and *N. bourbonica*. Note that except for the parts specifically labeled, the vertical lines in the distribution map represent cement roads with a width of approximately 3 m, and the size of the black solid circles in (**A**) corresponds to the actual size of the red imported fire ant nests.

**Figure 8 insects-15-00582-f008:**
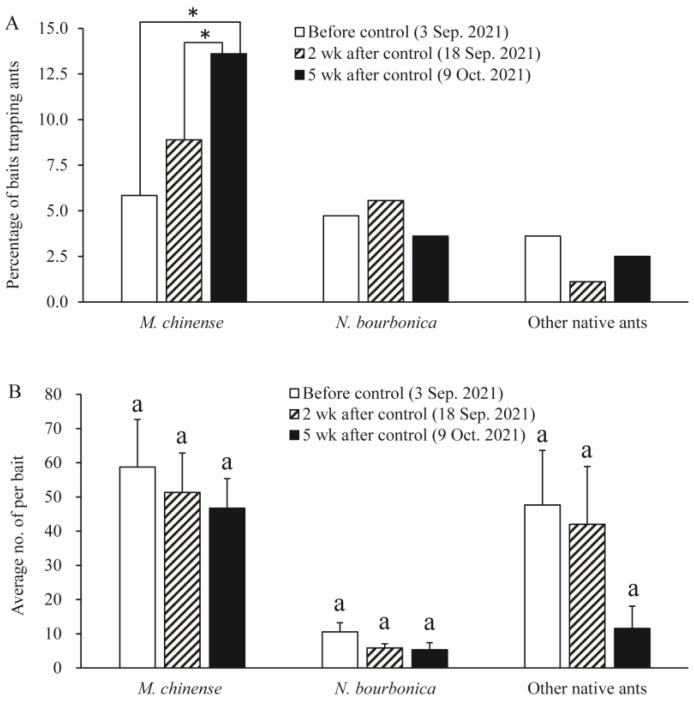
Abundance of native ants before control of *S. invicta* and 2 weeks and 5 weeks after control of *S. invicta* in the grasslands on the banks of the Dongyangjiang River in Jinghua, Zhejiang province. This was revealed by the proportions of baits trapping different ants (**A**) and the number of ants per bait (**B**). The asterisk indicates a significant difference between dates. Bars with same letter indicate no significant difference from each other (Tukey’s test, *p* > 0.05).

**Table 1 insects-15-00582-t001:** Scores (mean ± SE) for aggression observed in dyad interactions between native ants and *Solenopsis invicta* of different body sizes.

Native Ants	Large-Sized *S. invicta*	Medium-Sized *S. invicta*	Small-Sized *S. invicta*
*Monomorium chinense*	1.90 ± 0.34 ABb	2.70 ± 0.31 Aa	1.55 ± 0.31 Bb
*Nylanderia bourbonica*	3.45 ± 0.11 Aa	2.90 ± 0.19 Ba	2.55 ± 0.26 Ba
*Iridomyrmex anceps*	3.40 ± 0.11 Aa	2.95 ± 0.20 Aa	3.05 ± 0.20 Aa

Note: Values in the same column with different lowercase letters and those in the same row with different uppercase letters showed significant differences from each other (*p* < 0.05, Mann–Whitney *U* test).

**Table 2 insects-15-00582-t002:** Relations of the mortality rates of native ants and *Solenopsis invicta* at 36 h after treatment with the number of ants and species of native ant (factors) in group-on-group interaction tests.

Factors	Native Ants/ Ant Number	Mortality of Native Ants			Mortality of *S. invicta*		
		H	df	*p*	H	df	*p*
Ant number	*M.c.*	7.282	3	0.063	8.163	3	0.043 *
	*N.b.*	8.805	3	0.032 *	7.855	3	0.049 *
	*I.a.*	10.458	3	0.015 *	5.764	3	0.124
Native ant species	1 vs. 1	7.385	2	0.025 *	7.057	2	0.029 *
	3 vs. 3	7.200	2	0.027 *	7.261	2	0.027 *
	5 vs. 5	5.535	2	0.063	5.514	2	0.063
	10 vs. 10	6.713	2	0.035 *	7.322	2	0.026 *

Note: *M.c.*, *N.b.*, and *I.a.* stand for *Monomorium chinense*, *Nylanderia bourbonica*, and *Iridomyrmex anceps*, respectively. An asterisk stands for significance at the level of *p* = 0.05 (Kruskal–Wallis *H* test).

**Table 3 insects-15-00582-t003:** Relations of mortality rates of native ants and *Solenopsis invicta* at 36 h after treatment with the number of ants and species of native ants (factors) subjected to group-on-one interaction tests.

Factors	Native Ants/ Ant Number	Mortality of Native Ants			Mortality of *S. invicta*		
		H	df	*p*	H	df	*p*
Ant number	*M.c.*	8.568	3	0.036 *	7.267	3	0.064
	*N.b.*	5.647	2	0.059	6.557	2	0.038 *
	*I.a.*	5.744	2	0.057	0.000	2	1.000
Native ant species	1 vs. 1	7.385	2	0.025 *	7.057	2	0.029 *
	3 vs. 3	6.938	2	0.031 *	7.579	2	0.023 *
	5 vs. 5	3.006	2	0.223	7.322	2	0.026 *
	10 vs. 10	8.568	3	0.036 *	7.267	3	0.064

Note: *M.c.*, *N.b.*, and *I.a.* stand for *Monomorium chinense*, *Nylanderia bourbonica*, and *Iridomyrmex anceps*, respectively. An asterisk stands for significance at the level of *p* = 0.05 (Kruskal–Wallis *H* test).

**Table 4 insects-15-00582-t004:** Taxon composition and relative abundance of native ants collected from a location after the removal of *Solenopsis invicta* (Redwood Town and nearby areas, Longyou, Zhejiang, China, mid-September 2019 to late July 2022).

Ant Subfamily	Ant Species	% Individuals	% Baits Present with the Species	Average Individual Number per Bait
Myrmicinae	*Monomorium chinense*	41.1	27.9	88.7 (21)
	*Tetramorium caespitum*	14.5	12.3	93.2 (19)
	*Pheidole* sp. 1	7.7	5.0	91.9 (13)
	*Pheidole* sp. 2	0.86	0.69	107.1 (6)
	*Pristomyrmex punctatus*	0.06	0.06	-
	*Crematogaster* sp.	<0.01	0.06	-
Dolichoderinae	*Iridomyrmex anceps*	16.6	19.5	53.9 (13)
	*Ochetellus glaber*	7.8	9.4	58.3 (13)
Formicinae	*Nylanderia bourbonica*	10.9	24.8	29.8 (21)
	*Nylanderia flavipes*	0.35	0.19	-

**Table 5 insects-15-00582-t005:** Percentage of baits trapping various ants in some habitats of Zhejiang (ZJ) and Jiangxi (JX) provinces invaded by *S. invicta*.

Location	Habitat Type	Observation Date	Total No. of Baits Used	% Baits Trapping RIFA and Native Ants			
				*S. invicta*	*N. bourbonica*	*M. chinense*	Other Ants
Longyou port, ZJ (29.085 N, 119.265 E)	Grass plot	11 March 2022	92	66.3	0.0	1.1	15.2
		9 April 2022	89	85.4	0.0	4.5	10.1
Jinhua, ZJ (29.011–29.157 N, 118.007–119.822 E)	Grass plot II	2 September 2023	130	13.1	7.7	4.6	10.8
	Grass plot III	9 September 2023	26	7.7	11.5	0.0	23.1
	Grass plot IV	24 September 2023	30	30.0	6.7 (1)	3.3	3.3
	Grass plot V	24 September 2023	40	32.5	30.0 (2)	5.0	15
	Grass plot VI	22 October 2023	20	100	10.0 (2)	0.0	0.0
	Tree nursery	1 September 2023	110	10.0	6.4	2.7	27.3
Shangrao, JX (28.487 N, 118.007 E)	Lawn	14 October 2023	57	63.2	3.5 *	14.0 (1)	0.0
		22 October 2023	40	100	0.0	10.0 (4)	0.0

Note: The number in brackets indicates the number of baits trapping simultaneously the corresponding ant and *S. invicta* (occurring in the same vials). The asterisk indicates there was one bait that trapped *N. bourbonica* and *M. chinense* simultaneously.

## Data Availability

Data are contained within the article or Appendix A. The original contributions presented in the study are included in the article/Appendix A. Further inquiries can be directed to the corresponding author.

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
