# Peer review of "Discovering Native Ant Species with the Potential to Suppress Red Imported Fire Ants"

_insects, 2024, doi:10.3390/insects15080582_

Round 1

Reviewer 1 Report

Comments and Suggestions for Authors

The reviewed manuscript Ni et al. “Discovering native ant species with the potential to suppress red imported fire ants” (Insects 3130964) focuses on antagonistic interactions between native ants and S. invicta in eastern China, on the native ant abundance in various habitats, on their seasonal dynamics, and their spatial distribution in habitats invaded by S. invicta.

 This is an interesting study. The experimental design is good and was well executed.

 Although many studies, in China and elsewhere, have examined the effects of fire ants on native ant diversity, there is a general lack of empirical data on native ants that show aggression towards S. invicta and are able to coexist with this species. The study has the potential to provide solid data towards our understanding the interactions between native ants and S. invicta in areas outside its native range.

The paper is well-written and easy to follow. The introduction is concise, and the aims are clearly indicated.

The references are complete and, to my knowledge, include all relevant literature.

I have a small number of comments and suggestions regarding the study, which are outlined below and also within the body of the text.

General comments

·         It may be worthwhile to provide a brief summary of how S. invicta was controlled at your study sites, if such information is available. Alternatively, you can include references if they exist. This is important as it will provide information on non-target effects of this approach.

Key words

·         The key words should reflect the content of the paper and ideally not repeat words already appearing in the title (e.g., native ant species). Consider replacing.

Materials and Methods

·         L 112-115: Consider providing size of native taxa, as not all readers will be familiar.

·         L 134-136: You selected medium-sized S. invicta workers for the group-on-group and the group-on-one trials. Why? Please elaborate.

·         L 146-149: Why was the 10 on 1 combination used only for M. chinense? Mostly out of curiosity, but this information may also be of interest to others.

·         L 152-155: Providing a map of the study area will be nice but is not completely necessary.

·         L 188-191. These data are not mentioned elsewhere in the text. Would they be included in the paper, maybe as supplementary information?

Results

·         L 325-327: You state that “… the mortality rates of the native ants and S. invicta were correlated significantly…”, but it does not appear that you conducted correlation analyses. I would be careful with the wording. Same on L 351.

Discussion

·         L 515-516: No stinger in Formicinae, to which Nylanderia belongs.

·         L 543-545: Are you suggesting paying more attention (more research focus) to habitat margins or creating more margins? Not quite clear as presented. Of course, habitat fragmentation is known for its negative impact on great many animal groups, from insects to vertebrates. 

·         L 557-558: Are you suggesting supplementing existing control measures with the addition of native ants? Such an approach is never a good idea and has the potential to disrupt existing ant (and other) communities.

Comments on the Quality of English Language

Minor edits only.

Reviewer 2 Report

Comments and Suggestions for Authors

Fire ants (Solenopsis invicta) are one of the most problematic insect species in various parts of the world, causing various damage with high economic potential. One of the consequences of the invasion of these ants is the decline of part of the native fauna of other ants, with the consequent negative effect on the local community. Some studies suggest that some native ants do not respond passively to S. invicta invasion, but rather may confront the invader and eventually keep it at bay. This opens the door to potential natural control of the fire ant, given the high economic and ecological cost of using chemical insecticides. For this reason, this contribution is welcome, where the possible natural control of the fire ant by at least two other ant species, at least one native, is explored.

Pages 101 – 102 Three subspecies of Nylanderia bourbonica are known. Is there a possibility of knowing what the subspecies of this study is? Or is it not relevant? 

Dyad interactions. Beyond evaluation 4, was it observed whether the stinger (Solenopsis / Monomorium) or chemicals (Nylanderia, Iridomyrmex) of one of the ants caused death or discomfort in the other? Nylanderia and Iridomyrmex do not have a sting, their effect on Solenospis would be chemical, a little more difficult to evaluate.

Ant identification. Is it necessary to use genes to identify genera in ants? It seems to me not, not even for common species, although having sequences may be useful for other future studies. 

Pages 199, 239, 237 Remove italics from S. invicta

Pages 213 – 214 Was the local eradication of S. invicta with chemicals? This may be important because of its effects on native ants.

Data Analysis. As I am not an expert in ecology or statistics, I cannot evaluate this section or the conclusions derived from said methods.

Page 461 Place S. invicta in italics

Table Remove italics from sp in Crematogaster

Page 515 Nylanderia does not have a functional sting apparatus, which the authors refer to as stinging

Page 524 put author of gracilipes

Comment: In other parts of the world there are other Monomorium and Nylanderia interacting with S invicta, it remains to be found out if similar species can exert the same influence on S invicta, limiting their populations.

There is no acknowledgment section in general.

References. All 66 references are cited in the text
